# MiR-31 improves spinal cord injury in mice by promoting the migration of bone marrow mesenchymal stem cells

Yujuan Zhang[1], Lili Cao[1,2,3], Ruochen Du[1], Feng Tian[2], Xiao Li[1], Yitong Yuan[1]*, Chunfang Wang[1]*

1 Department of Laboratory Animal Center, Shanxi Medical University, Taiyuan, Shanxi, China,
2 Department of Key Laboratory of Oral Disease Prevention and New Materials, Taiyuan, Shanxi, China,
3 Department of Dental Medicine, Shanxi Medical University, Taiyuan, Shanxi, China

* wangchunfang@sxmu.edu.cn (CW); ytyuan@sxmu.edu.cn (YY)

## Abstract

### Background

Stem cell transplantation therapy is a potential approach for the repair of spinal cord injuries and other neurodegenerative diseases, but its effectiveness is hampered by the low rate of targeted migration of cells to the area of injury. The aim of this study was to investigate the effects of *miR-31* on the migration of bone marrow mesenchymal stem cells (BMSCs) and the regulation of *MMP-2* and *CXCR4* expression *in vitro* and *in vivo*.

### Methods

*eGFP*-expressing BMSCs were isolated and cultured for subsequent experiments. The experiments were divided into three groups: control group, *miR-31*agomir group, and *miR-31*antagomir group. Proliferation was analyzed using CCK-8 and flow cytometry; cell migration *in vitro* was analyzed using wound-healing and transwell assays. The mouse SCI model was prepared by the impact method, and cells were transplanted (3 groups, 12 per group). Relevant inflammatory factors were detected by ELISA. The BMS score was used to evaluate the functional recovery of the mouse spinal cord and the frozen section was used to analyze the cell migration ability *in vivo*. The *in vitro* and *in vivo* expression levels of *MMP-2* and *CXCR4* were evaluated by Western blot and immunohistochemical staining.

### Results

*In vitro* experiments showed that cells in the *miR-31*agomir group exhibited enhanced cell proliferation (*P*<0.05, *P*<0.001) and migration (*P*<0.001) and upregulated protein expression levels of *CXCR4* (*P*<0.01) and *MMP-2* (*P*<0.001) compared with cells in the control group. The results of *in vivo* experiments showed that the expression of pro-inflammatory factors was reduced after cell transplantation treatment. Cells in the *miR-31*agomir group showed enhanced cell-targeted migration ability (*P*<0.001), improved the function of damaged tissues (*P*<0.001), and upregulated *CXCR4* and *MMP-2* expression compared to the control group (*P*<0.001).

**Data Availability Statement:** All relevant data are within the paper and its Supporting Information files.

**Funding:** My research was supported by the following projects, National Natural Science Foundation of China(81371384, 82001326); Applied Basic Research Project of Shanxi Province (201901D211319); Key Laboratory Opening Project of Shanxi Province (KF2020-02).

**Competing interests:** The authors declare that they have no competing interests.

## Conclusion

Our experiment demonstrated that *miR-31* could promote the migration of BMSCs and *miR-31* could repair and improve the function of damaged tissues in SCI.

## Introduction

Spinal cord injury (SCI) is a serious and devastating neurodegenerative disease characterized by chronic pain, sudden loss of motor sensation, and associated oxidative stress and axonal degeneration, ultimately leading to lifelong disability [1,2]. Pro-inflammatory cytokines can make spinal cord injury neurological impairment more severe, so suppression of inflammation is another important principle in the treatment of spinal cord injury [3]. Globally, spinal cord injuries have a high morbidity and mortality rate and result in serious economic and social problems [4]. Therefore, the treatment of spinal cord injury has become a focus of scientific and medical research, which has promoted the development of several methods such as drug therapy, surgery, and cell therapy [5]. However, with the development of tissue engineering in recent years, stem cell transplantation is considered the most promising method, especially for bone marrow mesenchymal stem cells (BMSCs) [6,7]. BMSCs are stem cells with self-renewal and multidirectional differentiation as well as immunomodulatory properties [8]. At the same time, mesenchymal stem cells are very sensitive to changes in inflammatory conditions, and the inflammatory environment will regulate the anti-inflammatory potential of BMSCs [9]. In spinal cord injury, BMSCs are able to migrate to the region of injury to improve behavioral responses and differentiate into neurons for functional repair due to their low immunogenicity, targeted homing, and secretion of trophic factors that enhance neuronal survival. However, the limited ability to migrate to the area of injury limits the effectiveness and application of cell transplantation therapy [8,10,11].

MicroRNAs (miRNAs) are a conserved endogenous class of small, single-stranded, non-coding RNAs of approximately 22 nucleotides that inhibit the translation or stability of mRNAs by interacting with the 3'-untranslated (3'-UTR) regions of target genes and are involved in several biological processes [12,13]. In recent years, studies have found that miR-NAs play an important role in regulating apoptosis, proliferation, migration, and differentiation of BMSCs [13]. MicroRNA-31 (*miR-31*), one of the many reported miRNAs, plays an important regulatory role in biological processes such as cell proliferation and migration [14]. Studies have shown that *miR-31*, after activation, can act on *RAS/MAPK*, *Hippo*, *JAK-STAT3*, and other signaling pathways to regulate the proliferation and migration of the corresponding cells and thus promote the regeneration of skin tissue, intestinal epithelial tissue, skeletal muscle, and other damaged tissues [15,16]. Current studies have been focusing on the effects of *miR-31* on bone regeneration in mesenchymal stem cells. For example, in 2018 Jia et al. used gene chips to predict the regulation of mRNAs and miRNAs involved in osteogenic differentiation of ADSCs and found that *miR-31* was highly correlated with osteogenesis in ADSCs [17]. Similarly, Marupanthorn et al. found that ALP activity and osteogenic gene expression increased with increasing time during osteogenic differentiation in BMSCs treated with *miR-31* inhibitors, a strategy that may be used to promote bone regeneration [18]. In cancer, miR-31 is an important regulator, often acting as an oncogene involved in the proliferation, migration, and invasion of cancer cells [19]. However, there are multiple overlapping mechanisms in the characteristics of stem cells and cancer cells [20]. This research mainly studies the role of *miR-31* in stem cells. It has been previously shown that *miR-31* promotes the differentiation of

spinal cord-derived neural stem cells into neurons and *miR-31* has embryonic specificity in regulating the proliferation of neural stem cells [21]. However, the effect of *miR-31* on the migration of BMSCs is not clear. Therefore, it is important to investigate whether *miR-31* has a facilitative effect on the migration of bone marrow MSCs to repair damaged tissues.

## Materials and methods

### Animals

The enhanced green fluorescent protein (*eGFP*) transgenic mice (6–8 weeks, 15–20 g) model has been established by our group as previously reported. Female FVB mice (8–10 weeks, 18-25g) for SCI modeling were purchased from Spelford Biotechnology Ltd (Beijing China). The entire experiment was conducted in an enclosed environment. All mice were raised under standard conditions (temperature 23–26˚C, humidity 50–60%, normal 12 h light/12 h dark biorhythm) and were given free access to water and food. During the experiments, 3% sodium pentobarbital was used for anesthesia to reduce animal pain, and mice were euthanized after high-dose anesthesia at the time of sampling. All methods and operations of this experiment were performed in accordance with the regulations for the husbandry and use of laboratory animals at Shanxi Medical University, and approved by the Ethics Committee for Laboratory Animals of Shanxi Medical University (Approval No.: SYDL2021014).

### Isolation, culture and identification of BMSCs

BMSCs from the *eGFP* mice were extracted using the whole bone marrow apposition method. The *eGFP* mice were anesthetized intraperitoneally with 3% pentobarbital (30 mg/kg). Bilateral femurs and tibias were dissected after removing surrounding tissues and muscles, and rinsed with phosphate- buffered saline (PBS, Solarbio, China) for three times. Both ends of the bone were cut using tissue scissors, and following procedures were successively performed: (i) the bone marrow cavity was flushed three times using a 1 mL syringe; (ii) the culture medium containing bone marrow was collected; (iii) he supernatant was discard after centrifugation; (iv) the cell pellet was resuspended in DME/F-12 (Gibco, US); and (v) the resuspended cells were used to inoculate T25 culture flasks (Corning, US) to form a single cell layer. The cells were incubated at 37˚C in a 5% $CO_2$ cell incubator and then the medium was replaced with fresh medium 4 h later, and every three days thereafter. Cells would be passaged when they reached a confluency of 80%, and the 3rd to 5th passaged cells(P3-P5) were used in subsequent experiments. BMSCs were inoculated on six-well plates (Corning, US) and the expression of their important markers *CD90*, *CD34*, and *CD44* (BOSTER, China) were identified by immunofluorescence techniques.

### Cell transfection

P3-P5 BMSCs were inoculated into a six-well plate ($1 \times 10^5$ cells/well) containing fetal bovine serum (FBS, Gibco, US) medium. When the cells reached a confluency of 30–50%, the mixture of *miR-31*agomir (RIBBIO Guangzhou, China) and transfection reagent (Polyplus, France) and transfection reagent were separately added to the wells and the cells were further cultured for 12 h before the medium was replaced with normal medium. The final concentrations for *miR-31*agomir and *miR-31*antagomir were 50ηM and 100ηM, respectively.

### Transfection efficiency analysis

Since our customized *miR-31* siRNA drug is modified by Cy3 fluorescent labeling, the transfection can be directly observed under a fluorescence microscope. 24 h post-transfection, the

medium was discarded and the cells were then washed with PBS for three washes, fixed with 4% paraformaldehyde (Solarbio, China), and stained with HOCHEST (Germany). The experiment was repeated three times.

### Real-time fluorescence quantitative PCR (RT-qPCR)

Total RNA was extracted using TRIZOL reagent (TAKARA, Japan) then reverse transcribed into cDNA using a miRNA First-Strand Synthesis Kit (TAKARA, Japan). The expression of *miR-31* in cells after transfection was quantitatively determined using the TB Green kit (TAKARA, Japan) on a Real-Time PCR System. The primers used are shown in Table 1.

### Cell proliferation analysis

Analysis of *miR-31* on the proliferation of BMSCs was performed using cell counting kit-8 (CCK-8) (Solarbio, China). Briefly, BMSCs were inoculated in 96-well plates ($1.0 \times 10^4$ cells/well) and transfected when cell reached a confluency of 30–50%. Then 10 μL of CCK-8 solution was added to each well and the plate was incubated for 4 h before the absorbance value at 450ηm was measured using an microplate reader. The OD values at day 1, day 2, and day 3 were measured respectively, and 5 replicate wells were set in each group.

### Cell cycle analysis

Cell cycle was detected by flow cytometry. Briefly, $5 \times 10^5$ transfected cells were collected and analyzed using a cell cycle kit (Elabscience, China) by following the manufacturer's instructions.

### Cell migration analysis

Transwell experiment: The transfected cells were resuspended in a serum-free transwell (pore diameter: 8μm; Corning, USA) with 200μL ($1 \times 10^5$ cells) and 600μL of complete medium in the upper chamber and the lower chamber, respectively, and incubated for 24 h at 37˚C in 5% $CO_2$. The cells were then fixed in 4% paraformaldehyde and stained with 1% crystalline violet (Solarbio, China). Finally, images of five visual fields were collected for analysis under a microscope and the experiment was repeated three times.

Wound-healing assay: The cells were seeded in a six-well plate, transfected, and let grow to 100% confluency by overnight incubation. The next day, the cells were scratched vertically on the bottom of the dish with a yellow gun tip, washed slowly with PBS to remove the scratched cells, and incubated in serum-free medium in a 37˚C, 5% $CO_2$ incubator. Photographs were taken and images were collected at 0 h, 12 h, 24 h, and 48 h, respectively. The migration area was analyzed using Image J software. The calculation formula is: %migration area = (Ai-Al)/Ai, Ai denotes the area of the initial scratch; Al denotes the area of the scratch after incubation.

**Table 1. Primer sequence of q RT-PCR.**

| Primer | Sequence (5′-3′) |
| --- | --- |
| *miR-31* | AGGCAAGATGCTGGCATAGCTG |
| *U6* Forward | GGAACGATACAGAGAAGATTAGC |
| *U6* Reverse | TGGAACGCTTCACGAATTTGCG |

## Establishment of SCI model

The FVB mice were randomly divided into 3 groups with 12 mice in each group. The mice were fasted for 12 hours, anesthetized by i.p. injection with 3% sodium pentobarbital, fixed on the operating table, exposed the spinal cord, and hit to the spinal cord at T8-T10 with Impactor M-III (New York University). The spastic tail swinging, the retraction of the lower limbs, and the body of the mouse were used as signs of successful modeling, and the wound was sutured after the operation. To prevent post-operative infection, gentamicin (8mg/kg) was administered intramuscularly for 3 consecutive days and the mice were manually urinated twice daily until they resumed urinating on their own. Tissue from the injury site was extracted for histological examination on day 1, 3, 5, and 7 after modelling.

## ELISA of *IL-6* and *IL-17*

Blood samples from mice were collected and centrifuged at room temperature to separate sera samples for the detection of *IL-6* and *IL-17* using ELISA kits (BOSTER, China) according to the manufacturer's instructions.

## Cell transplantation and grouping

After successful establishment of the SCI model, cells were transplanted near the injury site at a rate of 1μL/min. Mice in control group, *miR-31*agomir group, and *miR-31*antagomir group were injected with 3 μL of serum-free medium containing BMSCS, 3μL of serum-free medium containing *miR-31*agomir-transfected BMSCS, and 3μL of serum-free medium containing *miR-31*antagomir-transfected BMSCS, respectively. Consistent number of cells was used in each group ($1.0 \times 10^5$/μL cells).

## Basso Mouse Scale (BMS) score

BMS scores were used after SCI model establishment and cell transplantation to evaluate the recovery of motor function in the hind limbs of mice. The mice were scored using a double-blind method to observe ankle joint movements, hind limb support of the hindquarters and forward stepping, and hind paw position and anterior and posterior limb coordination during stepping and trunk stability. The higher the score according to the BMS scale, the better the recovery.

## GFP immunofluorescence assay

Spinal cord tissue was fixed in 4% paraformaldehyde for more than 24 h and dehydrated with gradient concentration of sucrose. Sections of 8μm thickness were then prepared on a frozen section machine and finally observed under a fluorescent microscope and images were collected for analysis.

## Histological analysis

The tissue sections were stained with hematoxylin and eosin, and finally dehydrated and mounted.

The tissue sections were placed in citric acid antigen repair buffer (pH 6.0) for antigen repair. Next, endogenous peroxidase was blocked by incubationg with 3% hydrogen peroxide. The tissue section was blocked with 3% bovine serum albumin (BSA) and then incubated with primary antibody (*MMP-2, CXCR4*, BOSTER, China) overnight at 4˚C. The sections were washed several times and then incubated with secondary antibody (Absin, China) at room temperature for 50 min. The presence of specific antigens were developed using DAB

(BOSTER, China) reagent. The nuclei were double-stained after DAB development and observed under a microscope.

## Western blot analysis

Appropriate amount of RIPA (BOSTER, China) was added to the transfected cells and spinal cord tissues for 3–5 min to completely lyse the cells and tissues, and the supernatant was collected after centrifugation for the determination of total protein using the BCA protein concentration kit (BOSTER, China). The same amount of protein was separated by SDS-PAGE, and the resolved proteins were transferred to a PVDF membrane. The membrane was then blocked with 5% bovine serum albumin (BOSTER, China) for 1.5 hours, and sequentially incubated with primary antibodies overnight at 4˚C (*MMP2* antibody dilution ratio 1:10000, *CXCR4* antibody dilution ratio 1:1000, BOSTER, China), and second antibody (goat anti-rabbit IgG labelled with HRP, dilution 1:5000, absin, China) at room temperature. Western blot analyses were repeated for three times.

## Statistical analysis

We used the Student's T test to determine whether the data were normally distributed. Continuous data were expressed as mean ± standard deviation. One-way ANOVA and t-test were used for comparison between groups. All data were analyzed using SPSS 25.0 (IBM, NY, USA). Statistical graphs were produced by GraphPad Prism8 software (GraphPad, CA, USA), and a $P < 0.05$ was considered statistically significant.

## Result

### BMSCs cultured *in vitro* were *CD90*- and *CD44*-positive, but *CD34*-negative

BMSCs are easy to obtain and passage *in vitro*. Under an inverted microscope, the cells were consistent in morphology, resembling fibroblast-like cells that were spindle-shaped and in colonies (Fig 1B). Under the fluorescence inverted microscope, BMSCs showed green fluorescence (Fig 1C). Cellular immunofluorescence identification of BMSCs cells showed positive expression of *CD90* and *CD44* (98.4%,98.66%) and negative expression of *CD34* (0.14%) (Fig 1D and 1E).

### *miR-31* transfected BMSCs

We customized *miR-31*agomir and *miR-31*antagomir, which were modified by high affinity cholesterol and fluorescently labeled with Cy3. The red fluorescence was directly observed under fluorescence microscope after transfection of BMSCs (Fig 2C). The transfection efficiency was 93.27% in the *miR-31*agomir group and 94.44% in the *miR-31*antagomir group, with no statistically significant difference between the two groups (Fig 2D). The results of qRT-PCR experiments showed that after cell transfection, the expression of *miR-31* was significantly increased in the *miR-31*agomir group compared with the control group and was more than 7-fold higher ($P<0.001$), however, the expression of *miR-31* was restricted in the *miR-31*antagomir group ($P<0.01$) (Fig 2B). The cell morphology after transfection did not change (Fig 2E).

### *miR-31* promotes the proliferation of BMSCs

In order to study the effect of *miR-31* on the proliferation of BMSCs, we performed CCK-8 and Cell-cycle assays. As shown in Fig 3A, the proliferation of BMSCs was significantly

 

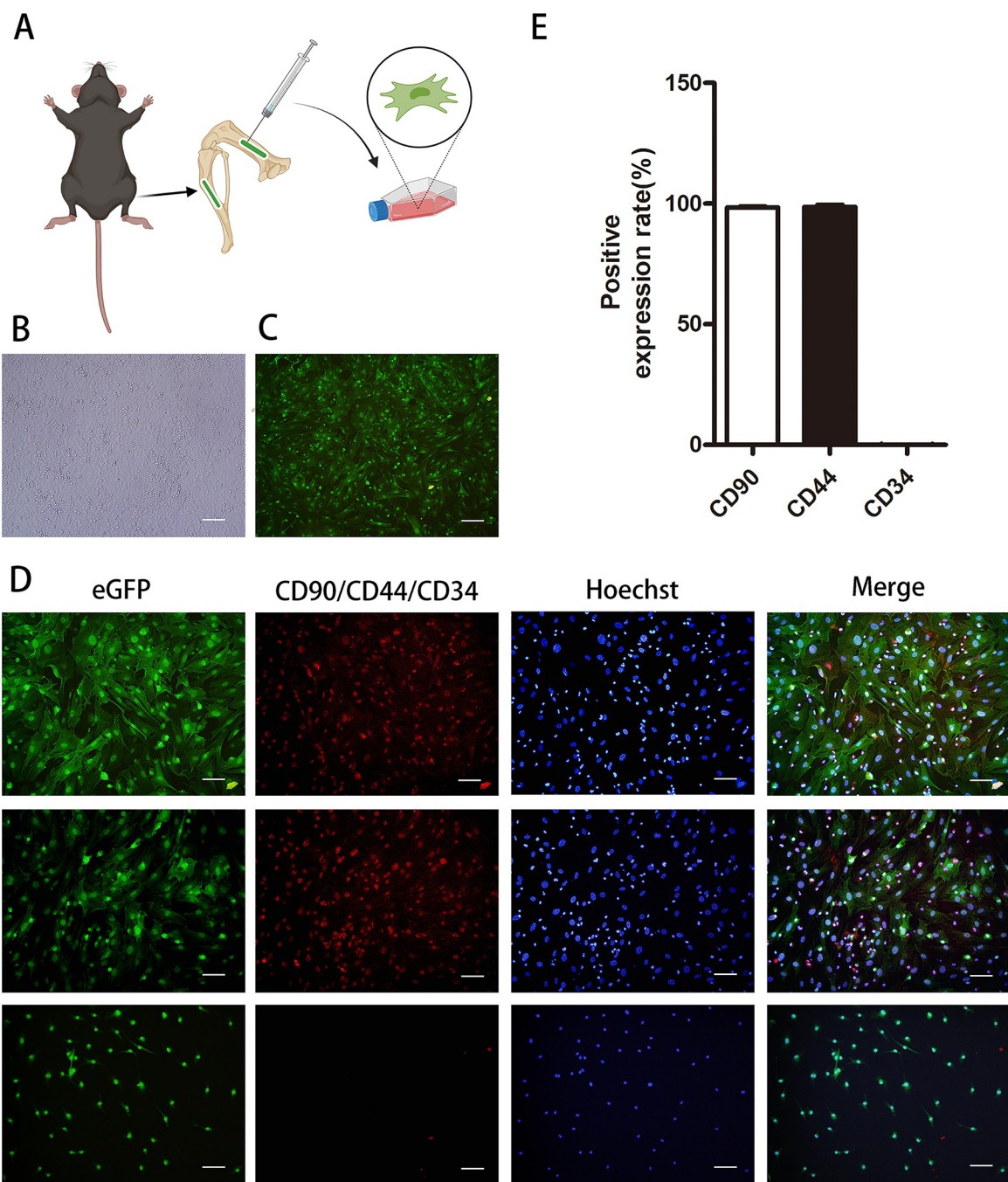

**Fig 1. Extraction and identification of green fluorescent protein-labeled bone marrow mesenchymal stem cells. A** Schematic diagram of cell extraction. **B** Cell morphology under an optical microscope (x100 magnification). Scar bar: 100μm. **C** Cell morphology under a fluorescent microscope (x100 magnification). Scar bar: 100μm. **D** The surface markers of BMSCs were identified by cellular immunofluorescence. *CD90* and *CD44* showed red fluorescence as antigen-positive expression; *CD34* did not show red fluorescence as antigen-negative expression. Scar bar: 50μm. **E** Statistical results on the expression of surface markers *CD90*, *CD44* and *CD34* in BMSCs. Dates are presented as mean ± SD (n = 5).

increased in the *miR-31*agomir group in a time-dependent manner compared with the control group (*P*<0.05, *P*<0.001). However, the proliferation of BMSCs in the *miR-31*antagomir group was inhibited at 72 h compared with the control group (*P*<0.001). Flow cytometry

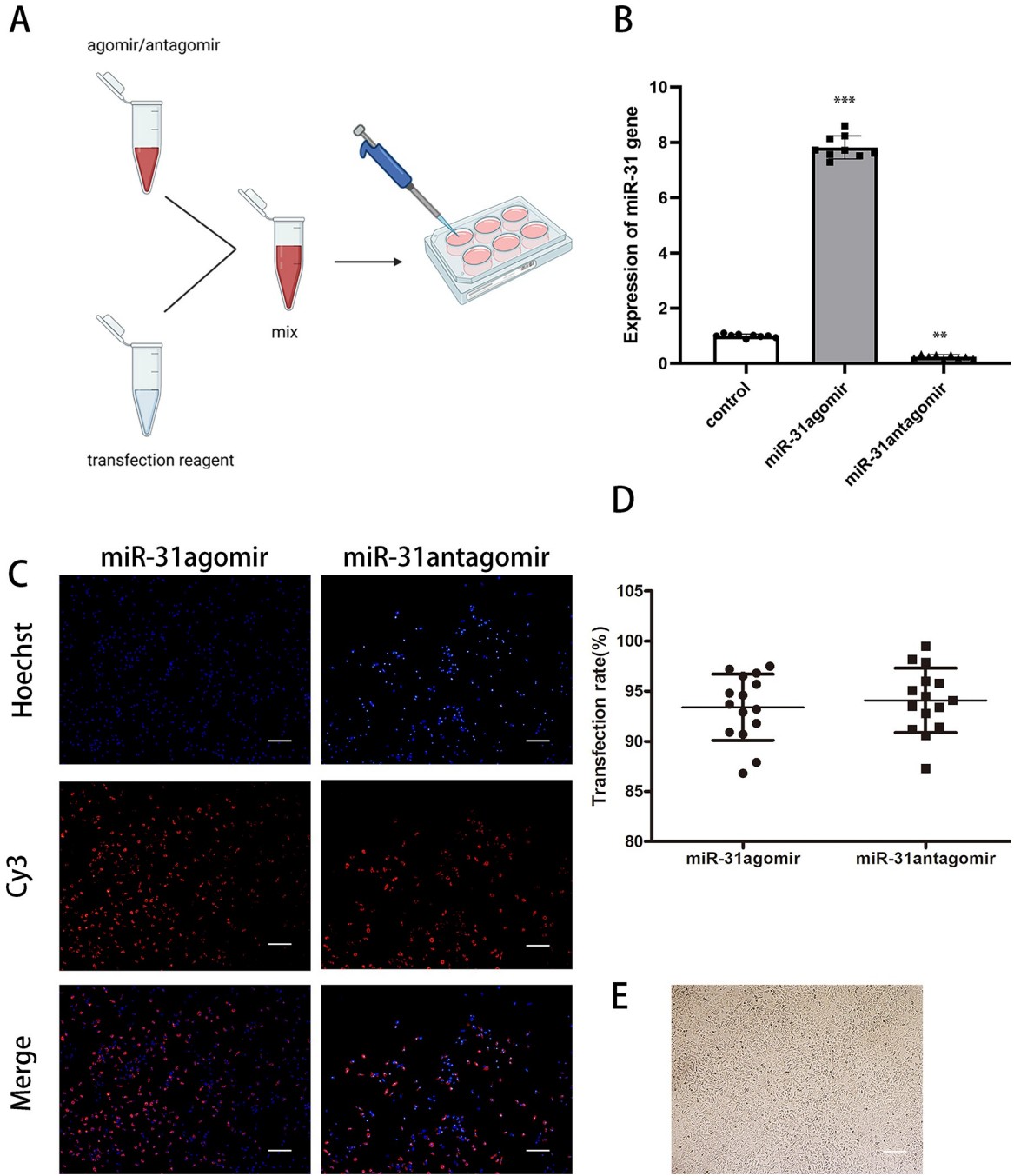

**Fig 2. *miR-31* transfection of BMSCs. A** Schematic diagram of cell transfection. **B** Expression of *miR-31* in bone marrow mesenchymal stem cells in each group after transfection was detected by qRT-PCR. **C** The status of *miR-31*agomir and *miR-31*antagomir transfection is detected by immunofluorescence, and successful transfection emits red fluorescence. Scar bar: 100μm. **D** Statistics of the results of *miR-31*agomir and *miR-31*antagomir transfection efficiency. **E** Morphology of the cells under optical microscopy after transfection. Scar bar: 100μm. Dates are presented as mean ± SD (n = 9,15). Statistical analysis: compared to control, **$P<0.01$, ***$P<0.001$.

results showed that the percentages of cells at the S phse in the *miR-31*agomir group, *miR-31*antagomir group, and the control group were 22.45%, 16.85%, and 19.68%, respectively (Fig 3B). In summary, *miR-31* can promote the proliferation of BMSCs.

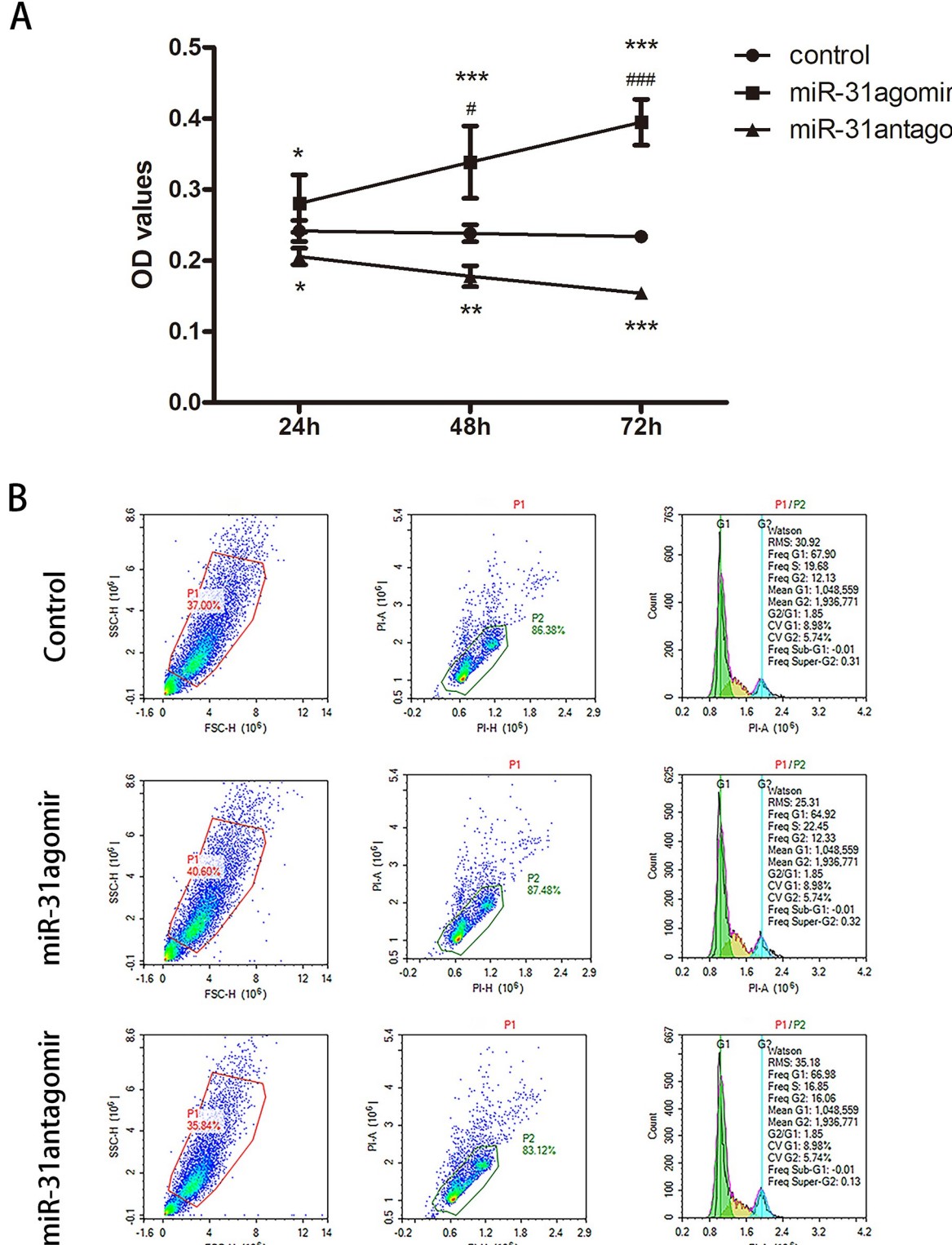

**Fig 3. Effect of *miR-31* on BMSCs proliferation. A** To analyze the effect of *miR-31* on the proliferation of BMSCs. The OD values of *miR-31*agomir, *miR-31*antagomir and control at 450m for 24, 48 and 72 hours were detected by CCK-8. **B** Detection of the cell cycle distribution of each group of cells by flow cytometry. Dates are presented as mean ± SD (n = 5). Statistical analysis: within group compared to control, *$P<0.05$, **$P<0.01$, ***$P<0.001$; between groups compared to 24 hours *miR-31*agomir, #$P<0.05$, ###$P<0.001$.

### miR-31 promotes the migration of BMSCs in vitro

To observe the effect of *miR-31* on the horizontal migration of cells, we analyzed the migration ability by wound-healing assay after transfection of the cells, which has been widely used to detect cell migration *in vitro*. At 24 h, compared with the control group, cells in the *miR-31*agomir group migrated faster ($P<0.01$). At 48 h, the cell migration rate was significantly faster in the *miR-31*agomir group than in the control group, with a significant difference ($P<0.001$). And the migration speed of cells in the *miR-31*agomir group accelerated with incresing time (Fig 4A). Statistical analysis of the wound healing area showed that the migration ability of BMSCs was significantly enhanced after transfection with *miR-31*agomir (Fig 4B).

In order to further confirm the effect of *miR-31* on the vertical migration of cells, a transwell chamber assay was used. The transfected cells were seeded in the upper chamber of transwell and cultured for 24 h (Fig 4C). As shown in Fig 4D, compared to the control group, more cells in the *miR-31*agomir group migrated to the lower chamber of the transwell. Additionally, as shown in Fig 4E quantitative analysis indicated that *miR-31*agomir could significantly enhance the migration of BMSCs ($p<0.001$). These experimental results suggested that *miR-31* could promote the migration of BMSCs, laying the foundation for repairing damaged tissue function.

### In vitro, miR-31 upregulats the expression of CXCR4/MMP-2 protein

To further investigate the mechanism by which *miR-31* promoted the migration of BMSCs, *CXCR4* and *MMP-2* were analyzed. Western blot results showed that the relative expression of *CXCR4* protein was higher in the *miR-31*agomir group ($P<0.01$) and lower in the *miR-31*antagomir group ($P<0.05$) compared to the control group (Fig 5B). Similarly, the relative expression of *MMP-2* protein and *CXCR4* expression showed the same trend (Fig 5C).

### Evaluation of miR-31 repair of spinal cord injury function by BMS score

To evaluate animal models of spinal cord injury, inflammatory factor assays, and hematoxylin-eosin staining experiments were performed. After the model was established, serum inflammatory factors were measured and the results showed that the relative levels of both *IL-6* and *IL-17* were higher in SCI mice than in normal mice ($P<0.05$, $P<0.001$) (Fig 6B). The BMS score was 0 (Fig 6C). As shown in Fig 6D, the normal spinal cord was morphologically intact with clear boundaries of gray and white matter, abundant neurons, and an intact central cord canal with no red blood cells, inflammatory cells or vacuoles. However, the morphological structure of the damaged spinal cord was missing, the boundaries of gray and white matter were blurred, the number of neurons was reduced, and erythrocytes, inflammatory cells, and vacuoles were scattered. These experiments indicated that the SCI model was successfully established.

After cell transplantation, the BMS scores were significantly higher in the *miR-31*agomir group compared to the control and *miR-31*antagomir groups, and the greatest significant differences were observed on day 5 and 7 ($P<0.001$) (Fig 6C). This result indicated that *miR-31* could promote the recovery of function in mice with SCI.

### In vivo, miR-31 can promote the migration of BMSCs

To evaluate the effect of *miR-31* on the migration of BMSCs *in vivo*, spinal cord tissues were selected for frozen sections on day 5 and day 7 after cell transplantation. As shown in Fig 6E and 6F, the mean fluorescence intensity in the tissue was higher in the *miR-31* group compared to the control group and *miR-31*antagomir group, and the fluorescence intensity was enhanced

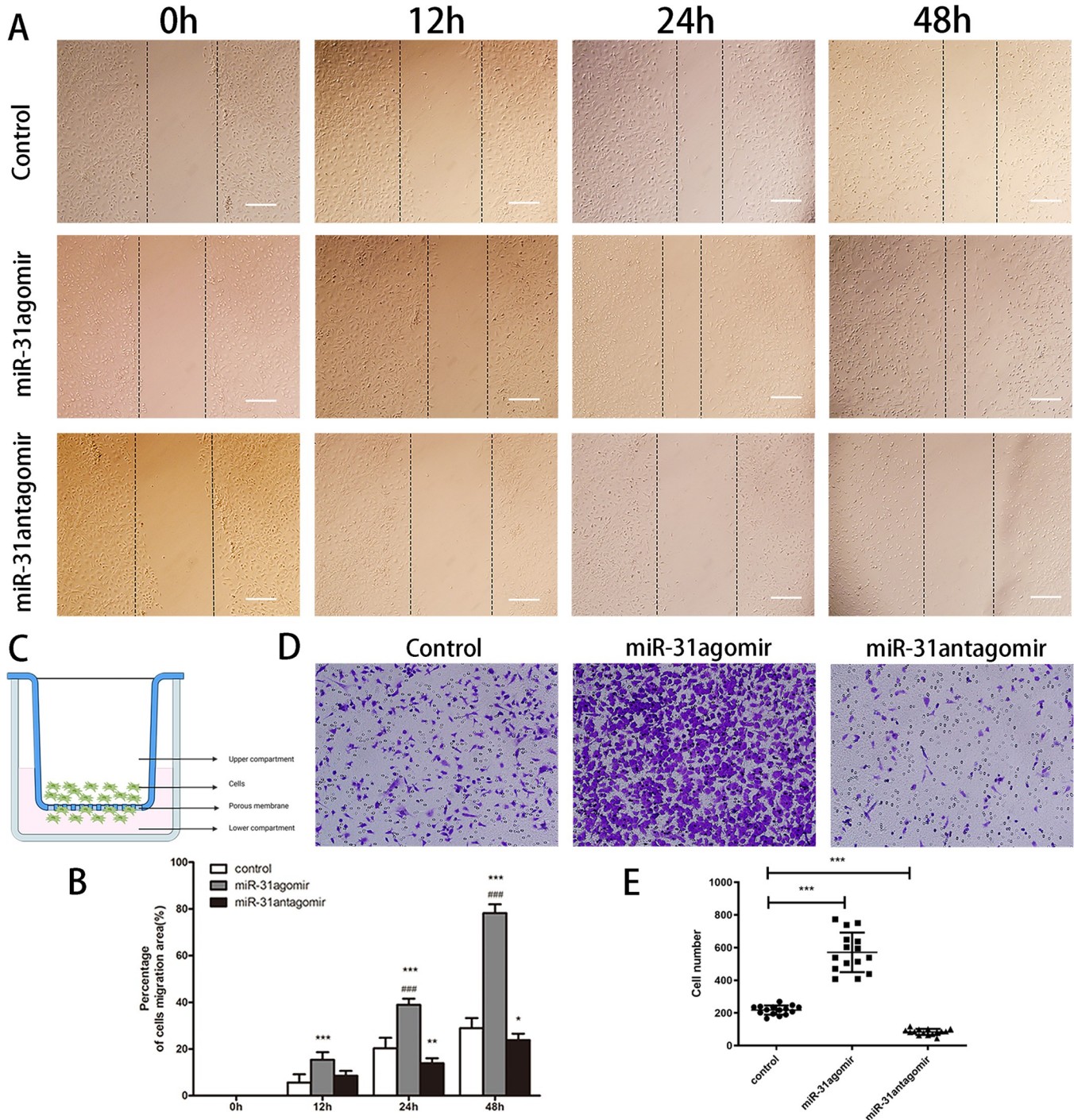

**Fig 4. Analysis of the effect of *miR-31* on the migration of BMSCs. A** The promoted effect of *miR-31* on the horizontal migration of BMSCs tested by bidirectional wound-healing assay at 0, 12, 24 and 48 hours. Scar bar: 100μm. **B** Percentage cell migration area result statistics for control, *miR-31*agomir and *miR-31*antagomir at 0, 12, 24 and 48 hours. **C** Schematic diagram of the vertical cell migration tested by Transwell chamber assay. **D** Cells that passed through the polycarbonate membrane were stained with crystal violet, observed under an inverted phase contrast microscope and counted in 200× magnification field. Scar bar: 50μm. **E** Statistical results of the number of cells in the lower chamber of the migration value of BMSCs. Dates are presented as mean ± SD (n = 5,15). Statistical analysis: within group compared to control, *$P<0.05$, **$P<0.01$, ***$P<0.001$; between groups compared to 12 hours *miR-31*agomir, ###$P<0.001$.

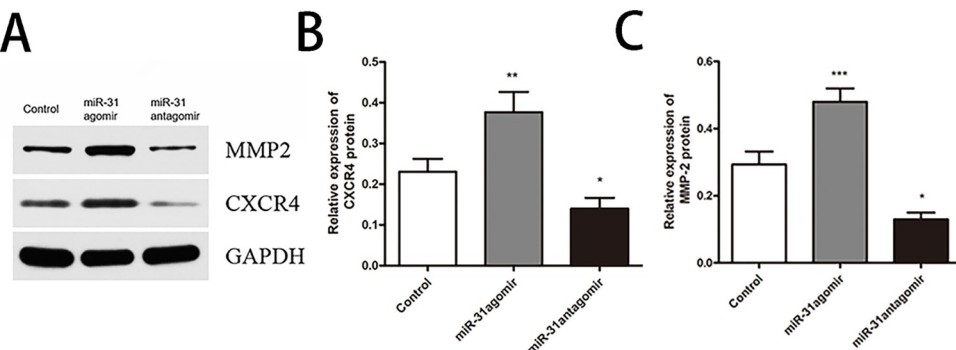

**Fig 5. In vitro, *miR-31* upregulated *CXCR4* and *MMP-2* expression. A** Strip chart of protein expression. **B** The relative expression of *CXCR4* protein in each group. **C** The relative expression of *MMP-2* protein in each group. Dates are presented as mean ± SD (n = 3). Statistical analysis: compared to control, $^*P<0.05$, $^{**}P<0.01$, $^{***}P<0.001$.

in a time-dependent manner ($P<0.001$). The results suggested that *in vivo*, *miR-31* had a facilitative effect on the migration of BMSCs.

## Cell therapy improves inflammation in SCI

To assess the expression of inflammatory factors after cell transplantation, we measured the relative levels of *IL-6* and *IL-17* on day 5 and 7. Compared with the control group, the relative expression levels of both *IL-6* and *IL-17* were reduced in the *miR-31*agomir group ($P<0.05$, $P<0.001$), with a decreasing trend with time (Fig 7E and 7F). The results indicated that pretreatment of BMSCs before transplantation would enhanced the anti-inflammatory effect in SCI.

## The *in vivo* effect of *miR-31* on the protein levels of *CXCR4* and *MMP-2*

To further investigate the mechanism of *miR-31* on the migration of BMSCs *in vivo*, we analyzed the expression of *CXCR4* and *MMP-2*. Immunohistochemical results showed a higher number of *CXCR4*-positive cells in the *miR-31*agomir group ($P<0.001$) compared to the control group, in a time-dependent manner ($P<0.001$) (Fig 7A and 7B). Similarly, the expression of *MMP-2* was consistent with the trend of *CXCR4*. Western blot results showed that the relative expression levels of *CXCR4* and *MMP-2* were higher in the *miR-31*agomir group ($P<0.05$) compared with the control group (Fig 7G). These experiments showed that *miR-31* upregulated the expression of *CXCR4* and *MMP-2* proteins.

## Discussion

Currently, BMSCs transplants are widely used in the treatment of SCI [22]. BMSCs promote functional repair after SCI by secreting various trophic factors and functionally useful growth factors, the biggest shortcoming of BMSCs transplantation is that they do not effectively migrate to the target tissues [23,24]. Hypoxia or trophic factors (stromal cell-derived factor-1 and monocyte chemotactic protein-1) have been shown to promote the migration of BMSCs *in vitro* [25,26]. However, there are few studies on the effect of *miR-31* on the migration of BMSCs. Therefore, in our study, cells were pretreated with *miR-31* siRNA to enhance the migration potential of BMSCs.

MicroRNAs (miRNAs) are small, non-coding RNA molecules that target mRNAs to silence gene expression at the post-transcriptional level and play an important role in a variety of cellular biological processes [12,27]. Research has shown that miRNAs used in the study of stem

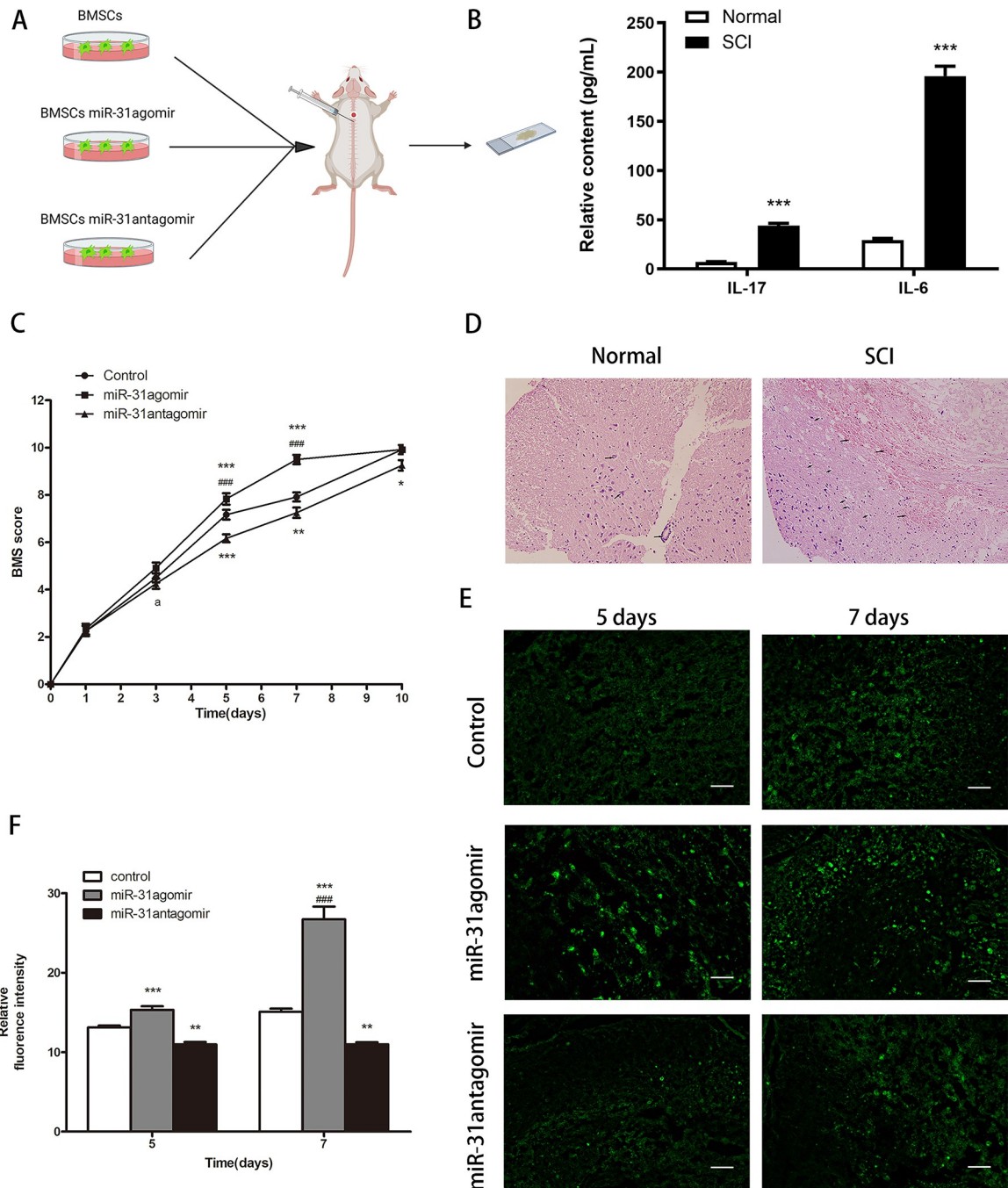

**Fig 6. Establishment of SCI model and the effect of *miR-31* on BMSCs migration in vivo. A** Schematic diagram of cell transplantation. **B** Relative levels of IL-6 and IL-17 in normal and spinal cord injured mice. **C** Assessment of functional repair after spinal cord injury in mice by BMS score. **D** The morphology of the normal and post-injured spinal cord was observed by HE staining. The arrow in the normal spinal cord points to the central cord and neurons; the arrow in the injured tissue points to the red blood cells and vacuoles. Scar bar: 50μm. **E** The migration of bone marrow mesenchymal stem cells within the spinal cord was observed by frozen sections on days 5 and 7. Scar bar: 50μm. **F** Statistical results of the relative fluorescence intensity of migrating BMSCs within the spinal cord. Dates are presented as mean ± SD (n = 5,6,12). Statistical analysis: Within group compared to control, $^{*}P<0.05$, $^{**}P<0.01$, $^{***}P<0.001$; between groups compared to 3,5 days *miR-31*agomir, $^{###}P<0.01$; $^{a}$ $P<0.05$ indicates *miR-31*antagomir compared to *miR-31*agomir.

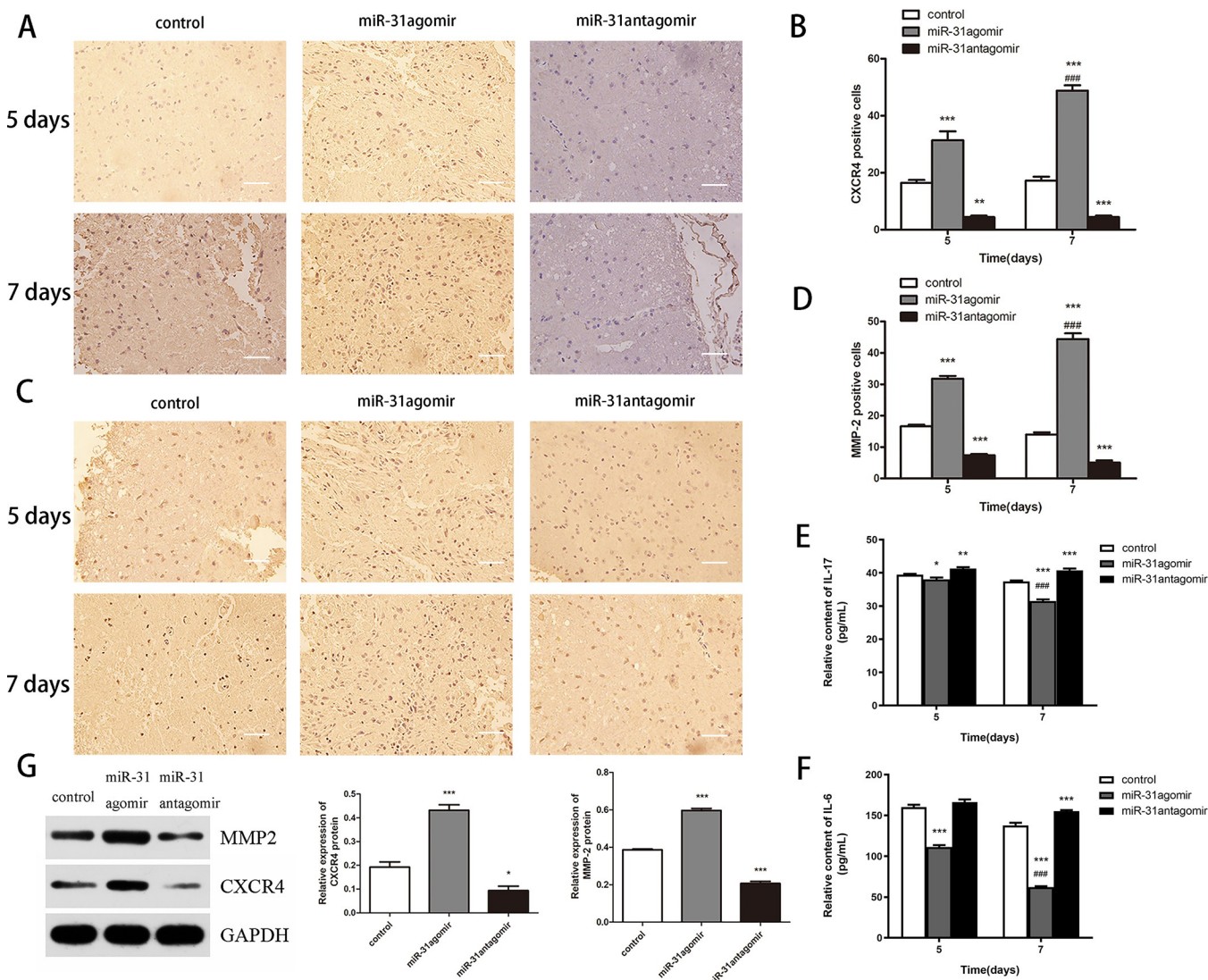

**Fig 7. *miR-31* upregulated the expression of *CXCR4* and *MMP-2*. A** After cell transplantation, *CXCR4* expression in tissues was detected by immunohistochemistry assays on days 5 and 7. Scar bar: 25μm. **B** Statistical results of *CXCR4* expression in tissues. **C** After cell transplantation, *MMP-2* expression in tissues was detected by immunohistochemistry assays on days 5 and 7. Scar bar: 25μm. **D** Statistical results of MMP-2 expression in tissues. **E** IL-17 expression in vivo after cell transplantation. **F** IL-6 expression in vivo after cell transplantation. **G** After cell transplantation, *CXCR4/MMP-2* protein expression was assessed in vivo by Western blot. Compared to control, $^*P<0.05$. Dates are presented as mean ± SD (n = 3,5,12). Statistical analysis: within group compared to control, $^*P<0.05$, $^{**}P<0.01$, $^{***}P<0.001$; between groups compared to 5 days *miR-31*agomir, $^{###}P<0.001$.

cells, tumor cells, and developmental regulation can regulate the expression of a wide range of genes and play a key role in the biological processes of stem cells [28–30]. For example, Satoshi et al. found that miR-720 promoted differentiation and reduced proliferation of dental pulp stem/progenitor cells by inhibiting the Nanog homologous frame (Nanog) [31]; Xiao et al. found that miR-146 played an important role in the regulation of stem cell proliferation, differentiation, and apoptosis [32]; Kong et al. found that microRNA-126 promoted proliferation and migration of bone marrow mesenchymal stem cells [33]. Studies have found that *miR-31* is involved in the biological process of cells, especially the invasion and migration of tumor cells [34]. However, there are limited studies that explore its effect on the migration of bone marrow mesenchymal stem cells.

*MiR-31* is highly conserved among species and it can be speculated that *miR-31* may play an important role in the proliferation and migration of BMSCs. In our study, *miR-31* siRNA was successfully transfected in BMSCs and the upregulation of *miR-31* expression in cells was confirmed by RT-qPCR. After comparing the migration ability of each group of BMSCs by transwell and cell scratch assays, *miR-31* was found to promote the migration of BMSCs. This result lays the foundation for efficient migration of BMSCs to target tissues. The migration process of BMSCs requires the synergistic action of several chemokines, adhesion factors, and proteases, and the most important of them are matrix metalloproteinase2 (*MMP-2*) and CXC chemokine receptor type 4 (*CXCR4*) [35]. Matrix metalloproteinases (MMPs) are a family of proteases that use zinc metal ions as cofactors and their main function is to degrade and maintain the homeostasis of the extracellular matrix [36]. Among them, *MMP-2* is the most important protein in the matrix metalloproteinase family and is also an important protein involved in cell migration. It plays a specific role in cell migration and proliferation by degrading collagen of the basement membrane [37]. The chemokine receptor family is the most tissue-specific and subpopulation-selective family of cell homing receptors, and of them, *CXCR4* has been shown to have a key role in injury-induced stem cell recruitment and a strong chemotactic role in regulating bone marrow MSC migration [38,39]. In this study, the results of Western blot experiments showed that the up-regulation of *miR-31* in BMSCs up-regulated the expression of *MMP-2* and *CXCR4* proteins, which is a prerequisite for efficient targeted migration after cell transplantation.

In recent years, it has been found that pretreatment of BMSCs prior to cell transplantation can promote the migration of BMSCs. Li-Ki et al. found that lithium preparations promoted the migration of BMSCs to restore neurological function in cerebral ischemia by inducing the expression of *CXCR4* in BMSCs [40]. However, current stem cell experiments do not give us information about the migration of transplanted cells in the target tissue [1]. Therefore, in this experiment, we chose to use green fluorescent protein expressing BMSCs pretreated with *miR-31* SiRNA for *in vivo* experiments. After the SCI model was successfully established, the migration of BMSCs in the damaged tissues and the expression of the associated migrating proteins were observed by collecting tissues for frozen sections and immunohistochemistry as well as Western blot detection. And the results showed that more BMSCs migrated into damaged tissues after pretreatment with *miR-31*agomir compared with those without pretreatment (*P*<0.01); the number of *MMP-2* and *CXCR4* positive cells and expression level of proteins in the *miR-31*agomir group were higher than in the control group (*P*<0.001). The results of *in vivo* experiments indicated that upregulation of *MMP-2* and *CXCR4* expression after cell transplantation was an important factor in improving the migration and homing of transplanted cells. Studies have shown that mesenchymal stem cells injected into mice can be located in the area of SCI. These cells can protect neurons, promote axon regeneration, and restore function to a certain extent [11]. Previous studies by our group have found that *miR-31* promotes functional improvement and repair after SCI in mice [21,41]. The prognosis of SCI and inflammation are closely related, and IL-6 and IL-17 are two important pro-inflammatory factors. IL-6 can cause early inflammatory response in SCI and enhance the expression of other inflammatory factors, and inhibition of IL-6 expression has a protective effect on SCI. Activated neutrophils after SCI are able to secrete IL-17, which greatly hinders the treatment of the disease, and the reduction of IL-17 levels can reduce the severity of SCI in the later stages. Therefore, we examined the relative levels of the two factors after animal model establishment and cell transplantation treatment, and the results showed that the levels of IL-6 and IL-17 were increased after injury and decreased after pretreatment transplantation of BMSCs, which suggested that pretreatment was more useful for disease recovery after cell transplantation. In this experiment, the results of BMS scores reconfirmed that cells modified by *miR-31*

could promote functional repair of damaged tissues. It may be that *miR-31* can inhibit the proliferation of glial cells after SCI. In conclusion, *miR-31* promoted the migration of BMSCs and upregulated the expression of *MMP-2* and *CXCR4*, which in turn promoted the repair after nerve injury.

This study specifies the effect of *miR-31* on the migration of BMSCs by *in vivo* and *in vitro* experiments. The strength of the current study lies in the *in vivo* experiments performed as well as the use of siRNA small molecule nucleic acid drugs. RNA interference (RNAi) is a natural cellular process that silences gene expression by promoting the degradation of mRNA [42]. The RNAi mechanism was discovered in 1998 by Fire and Mello in caenorhabditis elegans, but this mechanism was not further developed until 2006 [43]. Currently, siRNAs have become an important tool for RNAi to achieve gene silencing and can be used to regulate the expression of target genes by transfecting them into cells [44]. siRNA has also emerged as a potential and promising therapeutic platform and has been applied in many research areas. Conventional microRNA mimics or microRNA inhibitor is cumbersome to use and toxic to cells, which is not suitable for *in vivo* study and use in animals. Therefore, we attached a cholesterol moiety to the 3' end of *miR-31* small nucleic acid molecule, which can enhance its ability to cross the cell membrane and enter the cell to play a regulatory role, and has higher stability in animal experiments and cellular experiments [45]. Shu M et al. demonstrated that miR-335 transfected in glioma cells could be stably expressed in cells by PCR assay [46]. Hu JZ et al. showed that the efficacy of miR-21 on SCI was still detectable 4 weeks after direct injection at SCI, indicating that it could be stably expressed in animals [47]. Previous studies by our group have demonstrated that transfection of neural stem cells cultured *in vitro* with *miR-31* verified its stable expression in neural stem cells by PCR and effectively restored motor function after SCI within 21 days after injection into mice with spinal cord injury [47]. Due to the addition of a Cy3 light-emitting group, red fluorescence could be detected after transfection into the cells. Through this experiment, it was demonstrated that a large number of cells could emit red fluorescence after transfection with small nucleic acid molecules, and the results of fluorescence quantitative PCR showed that the expression of *miR-31* was significantly upregulated in the *miR-31*agomir group (*P*<0.001) and downregulated in the *miR-31*antagomir group compared with the control group (*P*<0.01), indicating that the synthesized small nucleic acid molecules could effectively regulate the expression of *miR-31* in BMSCs and thus determine the cell fate. And it was demonstrated that it could effectively restore the motor function of mice after SCI when injected with pretreated cells. Through the study, we found that the customized *miR-31* siRNA drugs exerted the same effects as common siRNA inhibitors and had a wide range of clinical applications. Thus, promoting the migration of BMSCs by *miR-31* provides a new therapeutic strategy for the treatment of neurodegenerative diseases such as SCI. It also lays a preliminary foundation for us to effectively use *miR-31*-related siRNA drugs for clinical treatment of SCI and motor neuron diseases.

## Conclusions

In this study, *in vitro* experiments demonstrated that at 48 h, 50ηM *miR-31*agomir significantly promoted the proliferation of BMSCs, showing a time-dependent increase compared to 24 h, and its effect was verified by cell cycle analysis. The time-dependent promotion of *miR-31* on the migration of BMSCs was demonstrated by migration assays. Thus, *miR-31* was shown to promote the proliferation and migration of BMSCs. In the *in vitro* experiments, the relative fluorescence intensity of cells in mice demonstrated that the cells had the strongest migration ability at day 5 and 7, and the efficacy of *miR-31* on the repair of motor function in SCI mice was demonstrated by BMS score.

Therefore, this experiment demonstrated that *miR-31* promoted the migration and proliferation of BMSCs by upregulating the expression of *MMP-2* and *CXCR4* to repair and improve the motor function of damaged tissues in SCI. However, whether it acts on *CXCR4/AKT* requires further studies to understand its detailed mechanism of action.

## Supporting information

**S1 File.**
(ZIP)

**S2 File.**
(ZIP)

## Acknowledgments

The authors would like to thank Zibo Yimore Translation CO. LTD for providing English proofreading services for this paper.

## Author Contributions

**Conceptualization:** Chunfang Wang.

**Formal analysis:** Yujuan Zhang.

**Funding acquisition:** Yitong Yuan, Chunfang Wang.

**Methodology:** Yujuan Zhang, Lili Cao, Ruochen Du, Yitong Yuan.

**Resources:** Yujuan Zhang.

**Software:** Lili Cao.

**Supervision:** Feng Tian, Xiao Li.

**Visualization:** Yujuan Zhang.

**Writing – original draft:** Yujuan Zhang.

**Writing – review & editing:** Ruochen Du, Yitong Yuan, Chunfang Wang.

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
