## [Decision Letter · Decision Letter 0]

25 Mar 2022

PONE-D-21-40503MiR-31 improves spinal cord injury in mice by promoting the migration of bone marrow mesenchymal stem cellsPLOS ONE

Thank you for submitting your manuscript to PLOS ONE. After careful consideration, we feel that it has merit but does not fully meet PLOS ONE’s publication criteria as it currently stands. Therefore, we invite you to submit a revised version of the manuscript that addresses the points raised during the review process.

We have struggled to attain reviews for this article. The single review we have suggests major revision, as below. As editor, I have read the manuscript and also have suggestions on improvement. Should you decide to progress with all suggestions, we will look at a revised article. 

1. The filed of miR-31 and MSCs is evidence for bone regeneration, however you do not refer to any of this research. Please widen your literature review of the activities of miR-31. 

2. Is the stability of miR-31 and the antagomiR verified in culture?

3. All images/figures are of a low resolution, this must be improved. 

4. the conclusion is too short and abrupt, this needs expansion and justification.

We look forward to receiving your revised manuscript.

Kind regards,

Catherine Berry, PhD, MSc, BSc

Academic Editor

PLOS ONE

Journal Requirements:

3. To comply with PLOS ONE submissions requirements, in your Methods section, please provide additional information on the animal research and ensure you have included details on (1) methods of sacrifice, (2) methods of anesthesia and/or analgesia, and (3) efforts to alleviate suffering.

5. Thank you for stating the following in the Acknowledgments Section of your manuscript: "The authors would like to thank the funders listed in the funding section for their support. The authors would like to thank Professor Chunfang Wang for helpful discussions.

Funding

National Natural Science Foundation of China(81371384， 82001326);

Applied Basic Research Project of Shanxi Province (201901D211319);

Key Laboratory Opening Project of Shanxi Province (KF2020-02)."

7. PLOS ONE now requires that authors provide the original uncropped and unadjusted images underlying all blot or gel results reported in a submission’s figures or Supporting Information files. This policy and the journal’s other requirements for blot/gel reporting and figure preparation are described in detail at https://journals.plos.org/plosone/s/figures#loc-blot-and-gel-reporting-requirements and https://journals.plos.org/plosone/s/figures#loc-preparing-figures-from-image-files. When you submit your revised manuscript, please ensure that your figures adhere fully to these guidelines and provide the original underlying images for all blot or gel data reported in your submission. See the following link for instructions on providing the original image data: https://journals.plos.org/plosone/s/figures#loc-original-images-for-blots-and-gels. 

8. PLOS requires an ORCID iD for the corresponding author in Editorial Manager on papers submitted after December 6th, 2016. Please ensure that you have an ORCID iD and that it is validated in Editorial Manager. To do this, go to ‘Update my Information’ (in the upper left-hand corner of the main menu), and click on the Fetch/Validate link next to the ORCID field. This will take you to the ORCID site and allow you to create a new iD or authenticate a pre-existing iD in Editorial Manager. Please see the following video for instructions on linking an ORCID iD to your Editorial Manager account: https://www.youtube.com/watch?v=_xcclfuvtxQ

9. Your ethics statement should only appear in the Methods section of your manuscript. If your ethics statement is written in any section besides the Methods, please delete it from any other section. 

Additional Editor Comments (if provided):

Thank you for your submission. Apologies regarding the delay in returning your paper to you, we have struggled to find sufficient reviewers for the article.

Reviewers' comments:

Reviewer's Responses to Questions

**Comments to the Author**

1. Is the manuscript technically sound, and do the data support the conclusions?

Reviewer #1: Yes

2. Has the statistical analysis been performed appropriately and rigorously? 

Reviewer #1: No

3. Have the authors made all data underlying the findings in their manuscript fully available?

Reviewer #1: Yes

4. Is the manuscript presented in an intelligible fashion and written in standard English?

Reviewer #1: No

5. Review Comments to the Author

Reviewer #1: In general, the paper theme is interesting and the methods are acceptable. However, The manuscript has several problems involving the space between the words that I indicated in the attached file with yellow color and with comments. Moreover, the statistical analyses and conclusions must be revised. I do not download the .tiff images, in the pdf, the quality of the images is poor.

6. PLOS authors have the option to publish the peer review history of their article (what does this mean?). If published, this will include your full peer review and any attached files.

Reviewer #1: No

---

## [Author Response · Author response to Decision Letter 0]

30 Apr 2022

1、The filed of miR-31 and MSCs is evidence for bone regeneration, however you do not refer to any of this research. Please widen your literature review of the activities of miR-31.

Thanks to the editors and reviewers for the questions I have answered as follows: MicroRNA-31 (miR-31), one of the many reported miRNAs, plays an important regulatory role in biological processes such as cell proliferation and migration. Studies have shown that miR-31, after activation, can act on RAS/MAPK, Hippo, JAK-STAT3, and other signaling pathways to regulate the proliferation and migration of the corresponding cells and thus promote the regeneration of skin tissue, intestinal epithelial tissue, skeletal muscle, and other damaged tissues. Current studies have been focusing on the effects of miR-31 on bone regeneration in mesenchymal stem cells. For example, in 2018 Jia et al. used gene chips to predict the regulation of mRNAs and miRNAs involved in osteogenic differentiation of ADSCs and found that miR-31 was highly correlated with osteogenesis in ADSCs. Similarly, Marupanthorn et al. found that ALP activity and osteogenic gene expression increased with increasing time during osteogenic differentiation in BMSCs treated with miR-31 inhibitors, a strategy that may be used to promote bone regeneration.

The revised section is on lines 70-83 of the article.

2、Is the stability of miR-31 and the antagomiR verified in culture?

Thanks to the editors and reviewers for the questions I have answered as follows: RNA interference (RNAi) is a natural cellular process that silences gene expression by promoting the degradation of mRNA. The RNAi mechanism was discovered in 1998 by Fire and Mello in caenorhabditis elegans, but this mechanism was not further developed until 2006. Currently, siRNAs have become an important tool for RNAi to achieve gene silencing and can be used to regulate the expression of target genes by transfecting them into cells. siRNA has also emerged as a potential and promising therapeutic platform and has been applied in many research areas. Conventional microRNA mimics or microRNA inhibitor is cumbersome to use and toxic to cells, which is not suitable for in vivo study and use in animals. Therefore, we attached a cholesterol moiety to the 3' end of miR-31 small nucleic acid molecule, which can enhance its ability to cross the cell membrane and enter the cell to play a regulatory role, and has higher stability in animal experiments and cellular experiments. Shu M et al. demonstrated that miR-335 transfected in glioma cells could be stably expressed in cells by PCR assay. Hu JZ et al. showed that the efficacy of miR-21 on SCI was still detectable 4 weeks after direct injection at SCI, indicating that it could be stably expressed in animals. Previous studies by our group have demonstrated that transfection of neural stem cells cultured in vitro with miR-31 verified its stable expression in neural stem cells by PCR and effectively restored motor function after SCI within 21 days after injection into mice with spinal cord injury. Due to the addition of a Cy3 light-emitting group, red fluorescence could be detected after transfection into the cells. Through this experiment, it was demonstrated that a large number of cells could emit red fluorescence after transfection with small nucleic acid molecules, and the results of fluorescence quantitative PCR showed that the expression of miR-31 was significantly upregulated in the miR-31agomir group (P�0.001) and downregulated in the miR-31antagomir group compared with the control group (P�0.01), indicating that the synthesized small nucleic acid molecules could effectively regulate the expression of miR-31 in BMSCs and thus determine the cell fate. And it was demonstrated that it could effectively restore the motor function of mice after SCI when injected with pretreated cells.

The revised section is on lines 482-511 of the article.

3、All images/figures are of a low resolution, this must be improved.

Thanks to the editors and reviewers for the questions I have answered as follows: All figures in the text have been revised and increased in resolution.

The revised figure has been submitted separately.

4、the conclusion is too short and abrupt, this needs expansion and justification.

Thanks to the editors and reviewers for the questions I have answered as follows: In this study, in vitro experiments demonstrated that at 48 h, 50�M miR-31agomir significantly promoted the proliferation of BMSCs, showing a time-dependent increase compared to 24 h, and its effect was verified by cell cycle analysis. The time-dependent promotion of miR-31 on the migration of BMSCs was demonstrated by migration assays. Thus, miR-31 was shown to promote the proliferation and migration of BMSCs. In the in vitro experiments, the relative fluorescence intensity of cells in mice demonstrated that the cells had the strongest migration ability at day 5 and 7, and the efficacy of miR-31 on the repair of motor function in SCI mice was demonstrated by BMS score.Therefore, this experiment demonstrated that miR-31 promoted the migration and proliferation of BMSCs by upregulating the expression of MMP-2 and CXCR4 to repair and improve the motor function of damaged tissues in SCI. However, whether it acts on CXCR4/AKT requires further studies to understand its detailed mechanism of action.

The revised section is on lines 518-530 of the article.

5、Avoid keywords with the words used in the title, this method decreases a future search by your paper. Use keywords related to the theme. 

Thanks to the editors and reviewers for the questions I have answered as follows: Key words: Marrow mesenchymal stem cells (MSCs); Anti-inflammation; miR-31; Cell migration; Spinal cord injury (SCI)

The revised section is on lines 41-42 of the article.

6、Let your text easy to the reader, for example: " (i) 1 mL syringe, (ii) aspirate the medium, (iii) collect the buffer, (iv) centrifuge and discard the supernatant, and (v) mix the precipitate". 

Thanks to the editors and reviewers for the questions I have answered as followsAnswer: Both ends of the bone were cut using tissue scissors, and following procedures were successively performed: (i) the bone marrow cavity was flushed three times using a 1 mL syringe; (ii) the culture medium containing bone marrow was collected; (iii) he supernatant was discard after centrifugation; (iv) the cell pellet was resuspended in DME/F-12 (Gibco, US); and (v) the resuspended cells were used to inoculate T25 culture flasks (Corning, US) to form a single cell layer.

The revised section is on lines 112-117 of the article.

7、You applied parametric tests, however, your data are parametric? Shapiro-Wilk was tested ？

Thanks to the editors and reviewers for the questions I have answered as follows: We used the Student's T test to determine whether the data were normally distributed. Continuous data were expressed as mean ± standard deviation. One-way ANOVA and t-test were used for comparison between groups. All data were analyzed using SPSS 25.0 (IBM, NY, USA). Statistical graphs were produced by GraphPad Prism8 software (GraphPad，CA，USA), and a P < 0.05 was considered statistically significant. 

The revised section is on lines 231-236 of the article.

8、To comply with PLOS ONE submissions requirements, in your Methods section, please provide additional information on the animal research and ensure you have included details on (1) methods of sacrifice, (2) methods of anesthesia and/or analgesia, and (3) efforts to alleviate suffering.

Thanks to the editors and reviewers for the questions I have answered as follows: During the experiments, 3% sodium pentobarbital was used for anesthesia to reduce animal pain, and mice were euthanized after high-dose anesthesia at the time of sampling.

The revised section is on lines 101-103 of the article.

---

## [Editor Report · Decision Letter 1]

21 Jul 2022

MiR-31 improves spinal cord injury in mice by promoting the migration of bone marrow mesenchymal stem cells

PONE-D-21-40503R1

Dear Dr. Chunfang Wang,

We’re pleased to inform you that your manuscript has been judged scientifically suitable for publication and will be formally accepted for publication once it meets all outstanding technical requirements.

Kind regards,

Catherine Berry, PhD, MSc, BSc

Academic Editor

PLOS ONE

Additional Editor Comments (optional):

The revised manuscript did take on board the suggestions given following the original submission by both the reviewer and the editor. All corrections were included and outlined clearly in the author's letter.

---

## [Editor Report · Acceptance letter]

26 Aug 2022

PONE-D-21-40503R1 

MiR-31 improves spinal cord injury in mice by promoting the migration of bone marrow mesenchymal stem cells 

Dear Dr. Wang:

I'm pleased to inform you that your manuscript has been deemed suitable for publication in PLOS ONE. Congratulations! Your manuscript is now with our production department. 

Kind regards, 

on behalf of

Dr. Catherine Berry 

Academic Editor

PLOS ONE